# Knowledge, Beliefs, and Experience Regarding Slow Lorises in Southern Thailand: Coexistence in a Developed Landscape

**DOI:** 10.3390/ani13203285

**Published:** 2023-10-21

**Authors:** Luke F. Quarles, Juthapathra Dechanupong, Nancy Gibson, K. A. I. Nekaris

**Affiliations:** 1Nocturnal Primate Research Group, Oxford Brookes University, Gibbs Building, Gipsy Ln, Headington, Oxford OX3 0BP, UK; 2Love Wildlife Foundation, Bangkok 10120, Thailand

**Keywords:** Thailand, Khao Lak, local ecological knowledge, slow loris, *Nycticebus*, conservation, coexistence

## Abstract

**Simple Summary:**

Deforestation is increasingly forcing primates into proximity with people. It is vitally important, especially for globally threatened species, that we understand how species navigate human-dominated environments and if these interactions incur threats to their populations. Local knowledge is a valuable source of information on underrepresented and cryptic species, and studies relating to nocturnal species are limited in Thailand, including those of the Endangered slow lorises (*Nycticebus bengalensis* and *N. coucang*). Here, we analyze data regarding the knowledge, beliefs, and experiences of local people from Khao Lak, in southern Thailand, an area that is composed of rural and urban areas. We conducted 36 interviews using photo cards to determine (1) where and how often people see slow lorises, (2) what they see them doing, and (3) what they know about them. We analyzed meaningful common words and how they clustered together. We identified 11 key concepts that define the local beliefs about slow lorises. We found that people believed that slow lorises ate fruit, were not aggressive, but were “bad omens”; they also reported that there used to be more lorises, but the interviewees did not specify how recently. We also determined that slow lorises were often seen in rural and urban areas and we posit that the people of Khao Lak live in passive coexistence with lorises with minimal conflict and exploitation. Lastly, we determined that electrocutions and road accidents are the principal threats to slow lorises in Khao Lak. We discuss how local knowledge can be a vital first step in developing conservation action plans for the species.

**Abstract:**

Natural landscapes are being converted for agriculture and other human uses across Asia and this development presents potential threats for specialist species of primates, like the Endangered slow lorises of Thailand (*Nycticebus bengalensis* and *N. coucang*). It is crucial to understand the interface between humans and slow lorises in order to conserve these primates. Local ecological knowledge provides valuable information about these cryptic nocturnal species. We conducted 36 semi-structured interviews in Khao Lak, in southern Thailand, asking (1) where slow lorises were seen, (2) what they were doing, (3) how often people see slow lorises, and (4) what people knew about them. We converted the interviews to free lists and determined the importance of key words. Our results show that the informants saw lorises frequently in six general locations: forests/trees (58%), electric wires (47%), towns/villages (36%), plantations (33%), homesteads (28%), and roads (17%). The most prominent places were forests/trees, wires, and plantations. Eleven key concepts defined local beliefs, with the most prominent being that lorises are awake at night (69%), eat fruit (42%), are not aggressive (25%), are bad omens (25%), and there used to be more of them (25%). Due to a minimal presence of persecution or exploitation of slow lorises by humans in the study area and general tolerance in the face of competition for the same resources and spaces, we suggest that, despite extensive persecution for the illegal photo prop trade in nearby Phuket, the people of Khao Lak seem to live in a state of passive coexistence with slow lorises. However, we did find some evidence that the pet and photo prop trade are still present in the area. It is crucial that coexistence be struck within the context of deforestation and urbanization so that conservation initiatives can take place within the community to further improve the lives of humans and the status of lorises.

## 1. Introduction

Forest loss is rapidly accelerating in southeast Asia, and this is especially true in global biodiversity hotspots [1,2]. Forest cover in Thailand has been declining steadily from about 70% in 1930 to 15% in 2005 [3]. Habitat loss is a dire threat to all wildlife in Thailand, and one of the main drivers of tropical deforestation is agriculture [1,4]. Monoculture plantations of oil palm and rubber are prominent across Thailand and contribute to continual deforestation which leads to high species loss [4,5,6,7]. Rubber plantations are mostly replacing natural ecosystems [8], presenting problems for species of primates that require native forests to thrive and have specialized diets. Due to the rapid expansion of agriculture in Thailand, it is essential to understand human–primate relationships for specific regions as land conversion forces more encounters.

Primates are globally imperiled with 93% facing active declines and ~68% at risk of extinction [9,10]. Threats such as habitat loss, along with direct loss from hunting and trapping, act as constant pressures eroding the foundations of healthy populations. As humans continue to fragment and reduce natural landscapes, they reduce the area in which primates can live and simultaneously increase the perimeter, or edge, of primate habitats, bringing interior species closer to the edge. “Edge effects” can give the impression of an abundance of rare species when they are simply being pushed to the perimeter. Edges affect communities through several means: they act as barriers for dispersal, inflict mortality, cause forced edge crossing, and bring about new interactions [11]. Human–primate interactions are complex and lie upon a spectrum from hostility to coexistence. For highly threatened species, it is crucial to understand how they persist in and navigate rapidly changing anthropogenic landscapes and what manner of human–primate interactions occur within these environments [12,13].

Slow and pygmy lorises (*Nycticebus* spp. and *Xanthonycticebus* spp.) are a group of nocturnal and cryptic primates that live in south and southeast Asia with ten recognized species, all of which are listed as threatened by the IUCN Red List [14,15,16]. Slow and pygmy lorises are imperiled due to a multitude of factors such as the illegal pet and photo prop trade, demand within traditional medicines and black magic, and direct consumption. Across a range of countries, such as India, Indonesia, Cambodia, Malaysia, and Vietnam, studies have revealed diverse sets of traditions, beliefs, myths, and taboos related to lorises that determine whether they are utilized as pets, hunted for food or medicine, or whether they are avoided entirely [17,18,19,20,21,22]. For instance, in some Sundanese communities of West Java, Indonesia, myths about slow loris (*N. javanicus*) blood causing all manner of phenomenon from the drying of soil and the dying of plants to landslides and the collapse of entire mountains seem to ensure that they are left alone [18]. In neighboring communities where beliefs such as these were absent, lorises were either eradicated or not as abundant. Thus, it is essential that local beliefs be fully cataloged in order to understand their impact on conservation efforts.

Traditional beliefs are diverse within countries such as Vietnam, Cambodia, and Indonesia and prevalent beliefs are that lorises are bad luck, have healing properties, and their blood and body parts can cure illnesses and injuries [17,18,21]. Lorises’ appearance and toxicity are potential reasons for established taboos against touching them or bringing them into one’s house in areas such as Java, Indonesia [18]. Slow and pygmy lorises are unique as the only group of primates and one of the only groups of mammals that produces venom. Loris venom is utilized primarily for intraspecific competition, creating wounds that can fester and become necrotic or cause death, and it also may be effective against ectoparasites [23]. Slow and pygmy loris venom has a broad range of effects on humans with the most severe reaction being death from anaphylactic shock [24]. 

Slow lorises are primarily exudativores, meaning that they mainly eat exudates including gum, sap, and phloem accessed by gouging on trees, and they rarely eat fruits [25]. Unlike other arboreal primates, lorises cannot jump and need continuous canopy to avoid traveling to the ground where they are susceptible to predation [26]. With all these specializations, they are presumed to have stringent habitat requirements, though lorises are increasingly found in more urban areas [27,28,29]. Thus, it is important to understand how these threatened primates navigate unnatural human-dominated landscapes.

Lorises have been found to inhabit disturbed areas in countries such as India and Indonesia, where they were found in and around areas that were inhabited by or disturbed by humans, such as logged forest, forest edges near villages, home gardens, tea gardens, agricultural gardens, and forest plantations [27,28,29,30]. Emerging evidence indicates that the lorises of Thailand may be able to persist within human-dominated environments such as eco-resorts, surrounded by highly developed areas [31]. Understanding where lorises can persist affects conservation translocations. Occasionally, organizations relocate lorises from urban to inappropriate forest areas—this is likely because these organizations wrongly believe that “a forest” is better for lorises [15,29,32]; such releases have resulted in high loris mortality.

There are two species of slow lorises native to Thailand: the Bengal (*N. bengalensis*) and the greater slow loris (*N. coucang*), which can be difficult to distinguish visually [33]. Both are classified by the IUCN Red List as Endangered, with populations decreasing, and have been protected under national Thai law since 1992 (the Wildlife Reservation and Protection Act, B.E. 2535). There is a general dearth of study on wild slow lorises in Thailand, save for some aspects of distribution as well as habitat preference and the densities of a few populations [28,34,35,36,37]. Additionally, little is documented in relation to local beliefs and knowledge on lorises in Thailand [19]. 

Local ecological knowledge (LEK) encompasses region-specific knowledge attained from lived experience within the local environment [38]. LEK aids in our understanding of distribution and behavior, especially for underrepresented taxa for which data on distribution and conservation are lacking. LEK has been used to gather information on several elusive species, including the Hispaniolan solenodon (*Solenodon paradoxus*) and the Hispaniolan hutia (*Plagiodontia aedium*), allowing researchers to assess their status as well as threats [38]. LEK also helps to elucidate the threats that the species face and can help determine attitudes towards the animals of focus, even with ~40 informants [20,39,40]. In past studies, LEK has provided crucial conservation information for slow lorises in Java, Borneo, and Cambodia [18,20,39]. Fundamentally, the perspectives and knowledge from local people help to inform effective conservation action [41,42,43], necessitating the incorporation of LEK into the baseline data collection for any little-known species.

To better understand the LEK surrounding slow lorises of Thailand, we conducted interviews with a local population in the south of Thailand. Here, utilizing an ethnoprimatological approach, we explore the knowledge, beliefs, and experiences of local southern Thai residents regarding slow lorises. Ethnoprimatology refers to the combination of primatological and anthropological approaches and “the viewing of humans and other primates as living in integrated and shared ecological and social spaces” [44]. Due to the exploratory nature of this study, we attempted to answer a broad array of research questions: (1) We were interested in how people interact with slow lorises, specifically if they utilize them as a resource (i.e., as food, in medicine, in magic, as pets, as photo props). (2) We wanted to know if lorises could live around human settlements. (3) We aimed to understand how often interactions occurred between people and lorises. (4) We wanted to understand people’s perception of lorises’ behavior and diet. (5) Importantly for their conservation, we wanted to catalogue what myths or taboos exist in the area surrounding slow lorises. (6) We wanted to gauge the knowledge people hold about lorises. (7) Finally, we wished to understand, through the interviews, what threats face slow lorises in the study area. 

## 2. Materials and Methods

We conducted interviews in Khao Lak, a series of villages along 20 km of beach, now tourist-oriented, in the Thai Mueang District of Phang Nga Province, Thailand. Outside of small, developed urban areas that support the tourism industry and local communities, Khao Lak is composed largely of palm oil and rubber plantations. Patches of undeveloped forest occur outside of the nearby Khao Lak Lam Ru National Park, but trends indicate that development is encroaching on those areas [4,5,7].

The interview team consisted of a foreign researcher who spoke only English and two native Thai speakers who relayed predetermined questions posed by the foreign researcher. We stopped at markets, landmarks, tea shops, and personal residences across four villages, as well as a national park, to conduct interviews. A total of 48 people (M = 31, F = 17) participated in 36 interviews. We conducted interviews across three sessions in May 2022, with interviews lasting approximately five minutes. Attempts were made to interview people individually but due to the close-knit nature of the community and the public setting, informants were joined by other community members during eight interviews. Participation was restricted to individuals who were born in the area or that were long-time residents of Khao Lak (ten or more years). Interviews, conducted in standard Thai, were informal and semi-structured.

Interviews consisted of a naming task and a question-and-answer session [13]. Upon completion, the translators repeated the key messages of the respondent to determine whether we had the information correct. With respondent permission, we recorded the interviews using a digital audio recording application (AVR, version 8.1.1) with the understanding that all responses remain anonymous; no participants declined to be recorded. Interviews were later transcribed by one of the authors who was a translator. For the interviews we followed the ethical guidelines proposed by the Association of Social Anthropologists of the United Kingdom and Commonwealth and that the University Research Ethics Committee of Oxford Brookes University approved.

For the naming task, we used a total of 16 photos (Appendix A), with 11 species that could exist in the area, ranging from common to uncommon as determined by IUCN status, behavior, and habitat preference (i.e., reported proximity to human landscapes). Additionally, we included one photo of *N. bengalensis*, known to range in the study area, and *N. coucang*, which is not confirmed to range in the study area [34,35]. Lastly, we included a picture of *X. intermedius* due to the potential they could be introduced into the area through release from illegal wildlife trade. We used A4 laminated sheets to display the images (Appendix A). We held up the pictures one at a time and asked the participant to identify all species and recorded how many they answered correctly, incorrectly, and did not know. Two decoy primate species from Africa (blue monkey, *Cercopithecus mitis*; common brown lemur, *Eulemur fulvus*) were included to validate the assumption that participants were indeed aware of the species in their area, rather than guessing [45]. The slow loris images were always shown last and, if the participant recognized the loris, they were asked to give its name, where they have seen it, what it was doing when they observed it, and how often they see it. Seven individuals were approached and interviewed that did not recognize any lorises. Lastly, we collected demographic details on age, sex, and occupation for potential future analysis that could address questions outside of the focus of this study. 

We converted the interviews into free lists for each question. We subsequently identified 11 key concepts that could define the cultural domain and documented the presence of these beliefs on a scale of zero (not mentioned) to two (mentioned with strong convictions) [18]. This resulted in a 11 × 36 matrix that we then put through a hierarchical clustering analysis in SPSS (version 28.0.0.0) using the Ward method and square Euclidean distance [43]. We produced a dendrogram from the results of the clustering analysis. We additionally identified the main areas lorises were reported to be seen and documented the presence of these beliefs on a scale of zero (not mentioned) to two (mentioned with strong conviction). We created a 5 × 36 matrix and subjected it to a hierarchical clustering analysis in SPSS with the same methods as the beliefs analysis. We created a second dendrogram to visualize the analysis results. 

We used salience to determine the prominence of certain experiences and beliefs. Salience was determined by calculating Smith’s S, where a higher value of S indicates higher salience [46]. Saliency is a method to demonstrate how meaningful a word is within a particular community of people, with words of higher saliency brought up more frequently and mentioned earlier in free lists. We generated saliency scores for the top 15 key words for each session [47]. We used Microsoft Excel (version 16.63.1) to determine the saliency of each word for the entire group by calculating the Smith’s saliency index for each open-ended question [46,48].

## 3. Results

### 3.1. Naming Task

For the naming task, informants (N = 48) answered correctly on average 59.6% of the time, incorrectly on average 2.8% of the time, and reported not knowing an average of 37.6% of the animals. Only the responses of individuals that recognized the loris were analyzed. Misidentified species can be found in Table 1. The decoy species, a common brown lemur (*Eulemur fulvus*), was misidentified in seven interviews and the blue monkey (*Cercopithecus mitis*) in two interviews (Table 1). It was never the case that both decoy species were misidentified during the same interview. The informants used four names to identify the slow loris: ling lom (“wind monkey”), ling phee (“ghost monkey”), ling jun (no proper translation), and nang eye (“shy Mrs.”). During two interviews, informants described the loris and its behaviors, but could not remember its name.

### 3.2. Question and Answer

#### Image Identification

During 34 interviews (94%), informants spontaneously reported that they see slow lorises that look like *N. coucang* and in 17 interviews (47%), they reported seeing lorises that looked like *N. bengalensis*. No individuals reported seeing *X. intermedius*. During 19 interviews (53%), informants reported seeing *N. coucang* but additionally indicated that they did not see the *N. bengalensis* in the area with some adamantly stating that the only one in the area was the “brown one”. A total of 15 informants distinguished between the two images based on color, distinguishing *N. bengalensis* as “white” and *N. coucang* as “brown”, “grey”, or in one case, “red”. 

For the cluster analysis of local knowledge and beliefs, we isolated 11 key concepts to define the cultural domain based upon common themes from the translated interviews (Table 2). Figure 1 shows the cluster analyses of where lorises were seen and local beliefs regarding lorises.

One informant (F 57) reported that a loris like *N. bengalensis* was used as a photo prop to attract tourists at the beach near their home starting five years prior and continuing until the start of the SARS-CoV pandemic, but the handler and the loris have not been seen since the start of the pandemic. Additionally, another (M 60) reported, in reference to both lorises, “some people touch them and take them to see the foreigners”. One other informant (M 65), referring to *N. bengalensis*, reported “the white one people keep in cages as pets”.

During nine interviews, informants reported that the slow loris was a “bad omen” or was “bad luck” and in four of the interviews, they specifically indicated that the older generation told them this. In two interviews, informants (M 51 and M 52) reported that they believed that it was good luck to see one, but neither offered further explanation. A reference to lorises possessing venom or “poison” was found in only four interviews (11%). One informant (F 42), a ranger for Khao Lak Lam Ru National Park, was bitten by a baby loris during a rescue, only receiving a rabies vaccination in that instance. She reported that the wound was not bad, akin to being bitten by a dog or cat. One respondent (F 34) shared a story of a loris entering the chicken coop at their homestead and eating eggs. She reported that they caught the loris in a shirt and it tried to bite them.

Figure 2 shows the areas where lorises were reported to be seen. In three interviews (8%), informants reported seeing a slow loris dead on the road or reported that they saw a loris get struck by a car.

Isolating each individual open-ended question, we found the words of highest salience for the 36 interviews. For each of the four questions, Figure 3 illustrates the most common and salient unique key words. The terms “poison” and “venom” did not have high salience.

## 4. Discussion

LEK provides an avenue to explore a wide array of questions that help to elucidate status, ecology, threats, and roadblocks to conservation success for cryptic animals [18,38,39,41,42,43,49,50]. In our study, LEK was able to reveal information about our target species on all topics that we pursued. (1) We were able to define several ways in which people interact with lorises in Khao Lak and our findings suggest that they did not exploit them for medicine or food and there was only one reported instance of a loris as a photo prop, before the SARS-CoV pandemic, and one report that people keep lorises as pets. Further research is needed to examine exploitation in the region. (2) We confirmed that lorises regularly range around human settlements and even discovered that they are utilizing rubber plantations regularly. (3) Though we did not get precise measurements of how often people see lorises, we found that people and lorises frequently interact. Although, reports from people indicate that they believe loris populations are declining. (4) Regarding LEK related to loris behavior, we found that people reported that lorises eat fruits and insects, but there was no mention of exudates in their diet. (5) Few myths and taboos emerged, but a prominent taboo was found against having a loris in or around one’s house. This taboo was linked with the myth that they are “bad omens”. Though, a small proportion of the informants believed lorises to be “good luck”. (6) Unexpectedly, LEK lacked general reference to loris venom, with few people reporting that lorises were toxic. (7) Lastly, LEK identified car strikes and electrocution as sources of mortality for lorises in Khao Lak. 

Where sites in different countries had diverse blood myths, beliefs about lorises possessing mystical powers, and taboos surrounding them, our informants in Khao Lak displayed only a few superstitious beliefs and taboos that highly overlapped with previous findings [18]. The idea of lorises as harbingers of misfortune appears pervasive as slow and slender lorises are also associated with bad omens across India, Sri Lanka, and Cambodia [17,22]. Though, In Khao Lak, there was a belief that lorises were bad omens, it does not seem to have developed into malice towards lorises and, from these reports, people do not proclaim to bother them, hunt them, or eat them locally. These findings are limited by the exploratory nature of the interview questions. Future research should attempt to design questions that focus on what others do with lorises, to eliminate any fear of self-incrimination and retaliation for reporting illegal acts.

The presence of the belief that lorises are good luck is contradictory to the majority of opinions but is not completely unheard of in other parts of their range too. The Iban people of Sarawak, Malaysia, also believed that it was good luck to see a loris and they should be undisturbed [20]. This is important when considering that few people were aware that lorises had venom or “poison”, and this fact was not even mentioned early in the interviews. This is contrary to studies across Indonesia and Malaysia, where superstitions were strong and venom was mentioned early on in interviews and at higher rates, with explanations of its effects, indicating its importance in the LEK of lorises [18,20]. Our results seem to indicate that locals believe that lorises are generally not a threat but would defend themselves if touched. Though severe bites have been catalogued in Thailand, a potential explanation for why venom knowledge is lacking in this area is that locals do not hunt, catch, or keep lorises as pets and, because of taboo, avoid instances where they could be bitten [18,51].

Informants reported seeing lorises at relatively regular intervals, indicating an undeniable pattern of overlap between the people and lorises. Additionally, lorises were frequently reported to be seen in human-dominated landscapes. Indeed, one of the most salient words was “around”, indicating that lorises were within areas where people spend their time, reinforcing the idea that lorises are living within the human landscape. This finding highlights a trend that highly specialized and threatened animals live in and around human-dominated landscapes, strengthening the argument that they should not be translocated away from these areas [29,52,53,54]. However, it must be noted that edge effects may have a role to play in these findings. Regardless, future conservation efforts should focus on how to restore degraded human-dominated areas to accommodate the animals that exist there [55]. 

Respondents repeatedly mentioned seeing lorises in “plantations”, often specifically mentioning rubber plantations. This finding is corroborated by observations made of *N. bengalensis* in rubber plantations in China as well as in Thailand where *N. bengalensis* utilized new and old palm and tamarind plantations [28,40]. Future research should focus on how slow lorises and other species use rubber plantations and in what capacity they can use this disturbed monoculture habitat. As much of southeast Asia has been assessed to be a lose–lose area for rubber expansion, with high extinction vulnerability and poor rubber suitability, future studies could focus on collaborating with plantation owners to attempt to accommodate the presence of wildlife through agroforestry, creating mutual benefits [56,57,58]. In agroforestry landscapes in Java, farmers that grow coffee also plant a variety of crops that provide multiple layers of vegetation that support numerous species, including *N. javanicus,* and provide sustainable products for the farmers [57]. In North Sumatra, the implementation of natural fencing by planting suren trees (*Toona sureni*) in agroforestry landscapes secured the soil, protecting human areas from landslides, and creating habitat for species that controlled pests. Additionally, the added trees provided wood and leaves that yielded additional income for the farmers [59,60]. These efforts provide a framework for the implementation of diverse agroforestry projects that might provide shared benefits. 

When respondents spoke of loris feeding ecology, they most often claimed that lorises ate fruit (~39%) but only reported them eating insects in a few interviews (~14%). A study of local beliefs in Cambodia regarding *X. intermedius* showed similar findings, where locals reported lorises eating insects (36%) and fruits (26%) [17]. LEK regarding feeding ecology has been a helpful supplement to field study for elucidating the habitats of nocturnal and elusive species like forest elephants (*Loxodonta cyclotis*), however LEK is not always free from inaccuracies [50,61,62] and this might be especially true for cryptic nocturnal species. With many reports of lorises eating fruits, further research is needed to assess which fruits and what nutrition they contain for lorises. As was the inference in a brief study at an eco-resort in Khao Lak, it could be that they are consuming the latex from rotting fruits to supplement their diets [31]. Alternatively, people could be observing lorises feeding on insects that gather in fruit trees. In neighboring Vietnam, *X. intermedius* were considered a fruit-eating crop pest, but were theorized to be eating insects, the true pest species [63]. *N. javanicus* have been recognized in Java, Indonesia, as a valuable form of pest control, eating insects that feed on agricultural products [56]. If slow lorises are linked with this pest-control function in Thailand, it could have positive implications for their conservation within the agricultural landscape if agroforestry projects are initiated. At the same time, it is crucial to address the lack of knowledge regarding feeding ecology which could hinder translocation success if unsuitable areas are selected and can cause problems in captive settings where lorises are often fed diets high in fruits that cause dental disease [64]. This gap in knowledge could be addressed through educational programs with schools and conservation organizations within Thailand [65,66]. 

Lorises face threats within human-dominated landscapes related to their attempts to use human infrastructure. Several informants reported that there used to be more lorises or that there are fewer individuals now that the town, village, and market areas are no longer forested. This finding tracks with our understanding that local people appear to be sensitive to population declines and often link them either overtly or covertly to a rationale [17,67,68]. Local people appeared to link loris population declines covertly with electrocution. Additionally, these data indicate that the belief that there were fewer lorises is tied with not touching them.

When respondents indicated that there were fewer lorises left, they would often explain that they do not bother the lorises or touch them. We interpret this to have developed to prevent injury that is now a mindset that sustains passive preservation in Khao Lak. Lorises were also reported to cross roads in a small proportion of interviews with reports of lorises being struck by vehicles in 8% of interviews. Though lorises generally prefer continuous canopy, they will use electric wires and on rare occasions will walk on the ground to travel [56,69,70]. Across various areas within India and Indonesia, electrocution and road accidents have been recorded as a source of mortality, identified in some areas as being just behind habitat loss as the major reasons for population declines [32,51,69,70,71]. Road strikes and electrocutions are a problem for countless species of primates living in developed landscapes and should be addressed in future conservation efforts through an increase in canopy bridges, canopy trimming, insulation of power lines, and the creation of physical barriers on lines and posts to prevent use [72]. 

While prevention of mortality in urban areas and attempting agroforestry campaigns within plantations are promising steps to conserving lorises, providing sustainable habitat to connect to is crucial to improving the state of lorises within the region [30]. With increasing deforestation and edge effects forcing lorises into human-dominated landscapes, attempts to create more protected habitat for lorises may be crucial. Historical attempts to legally protect public lands, such as national parks, have not guaranteed habitat preservation. In fact, practices such as illegal logging, poaching, and agricultural encroachment have been tolerated and even encouraged by local authorities in some regions [73]. While efforts to protect public lands have had varied results, privately protected areas display a growing potential to benefit primate conservation efforts. A study of the Khao Lak Merlin resort in the study area was found to provide suitable habitat for family unit of lorises (*N. bengalensis*) [31]. These results indicate that more private landowners in Khao Lak could follow the model of the resort to benefit lorises. Indeed, evidence indicates that eco-resorts may be a source of private habitat protection and restoration. Eco-resorts have displayed a potential to assist habitat conservation through forest protection, restoration, and the implementation of no-take zones within coastal reef ecosystems [74,75]. In countries like Thailand, where ecotourism comprises a large part of the national economy, integrating them into conservation management plans could be a positive step.

Our study is potentially limited by its relatively small but not entirely uncommon sample size of 48 participates [39,40], and of course we see these data as a baseline that can be explored further. Still, we believe that our sampling covered a diverse array of areas within Khao Lak, and our data represent a relatively holistic view of LEK within the area. The questions were straightforward and the answers we received were relatively consistent. Additionally, there is always a concern that responses could have been influenced by the presence of a foreigner [76], though this is a relatively standard way of performing studies of this type and questions were not of a sensitive nature. Furthermore, interviews were designed to be largely carried out by the native Thai speakers and we associated ourselves with employees of a relatively well-known organization, the Khao Lak Merlin Resort.

## 5. Conclusions

Our findings provide valuable opportunities for local conservation organizations to target gaps in education, tackle conservation issues, and improve the human–loris interface, especially in agricultural settings. The LEK we were able to gather surrounding lorises has provided invaluable insights into the human–loris interface in Thailand. Crucially, the information from informants has helped us to understand that lorises are reported not to be exploited in Khao Lak in the same ways that they are in other neighboring countries and even within other areas of Thailand. Due to a general lack of reports of persecution or exploitation of lorises and general tolerance in the face of competition for the same spaces, and to a lesser extent resources, we can make a preliminary suggestion that the people of Khao Lak live in a state of passive coexistence with lorises. Future research is needed to understand the size of the loris population and the extent of exploitation in the region. Their presence seems to lend credence to the notion that lorises do not require pristine habitats to thrive and that they regularly live in developed places. Conservation efforts within Thailand should focus on how to prevent conflict and mortality in instances of overlap between people and lorises and promote mutually beneficial changes to agricultural and urban landscapes. Future studies should focus on expanding on the topics identified in this study. Further exploration into the exploitation of lorises in the region is needed, as well as an analysis of the diet of lorises in urban areas with a special focus on fruit consumption, and lastly, more information is needed on how lorises are utilizing rubber plantations and other monoculture agricultural landscapes.

## Figures and Tables

**Figure 1 animals-13-03285-f001:**
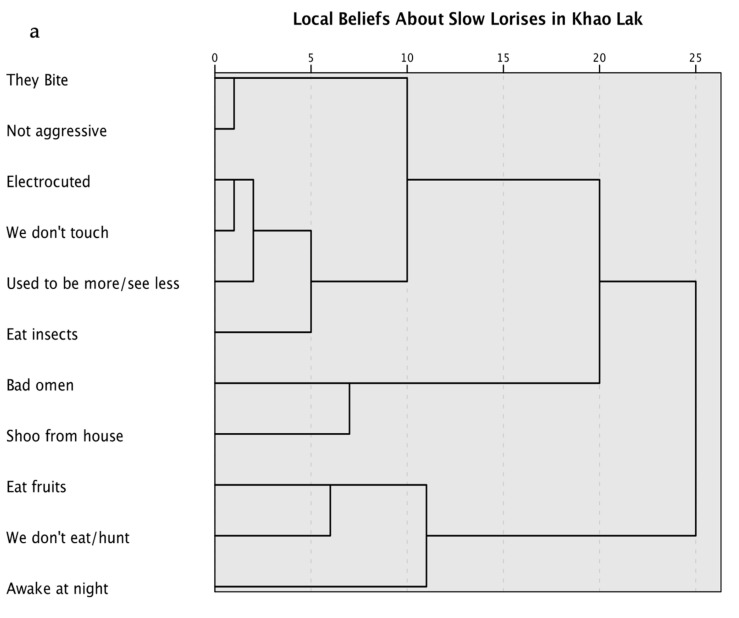
Cluster analysis of Khao Lak residents’ beliefs and experiences with slow lorises *Nycticebus bengalensis* in Khao Lak, Thailand based on the views of 36 informants. (**a**) clustering based on topics showing 3 or 5 potential clusters each comprising between 2 and 6 topics; (**b**) clustering based on where informants saw lorises, showing 3 distinct clusters each comprising between 1 and 2 locations each. Proximity matrices and agglomeration schedules in Appendix B.

**Figure 2 animals-13-03285-f002:**
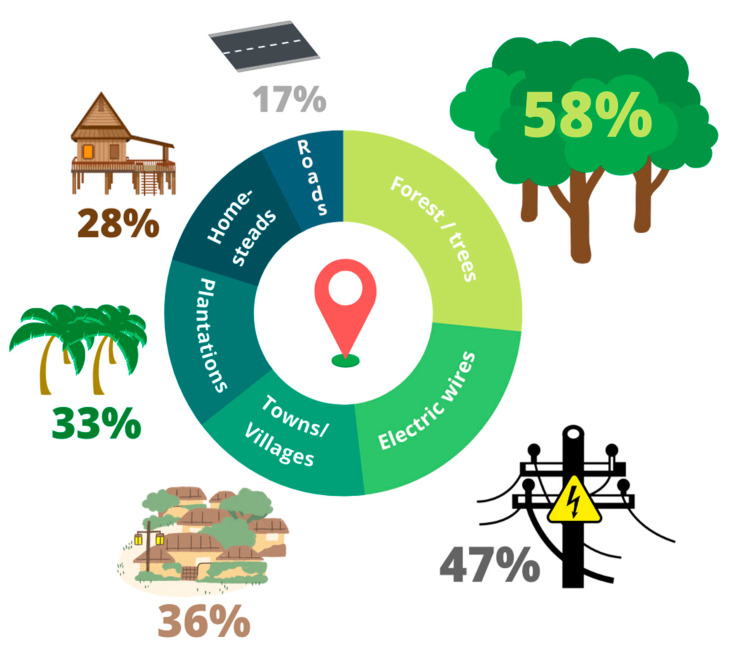
Locations where slow lorises were reported to be seen by interview informants. Representing the responses from 36 interviews around Khao Lak, Thailand.

**Figure 3 animals-13-03285-f003:**
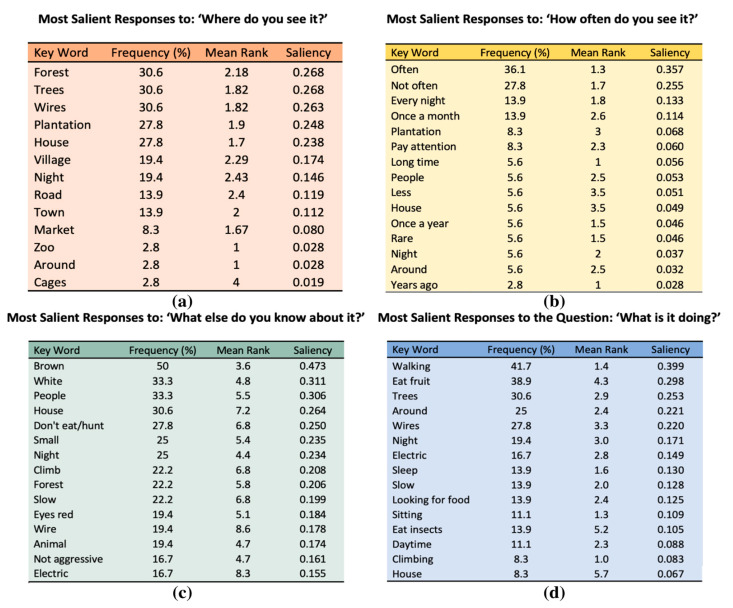
Frequencies, mean rank, and saliencies of the top key words (up to 15) describing slow lorises for the questions (**a**) “Where do you see it?”, (**b**) “How often do you see it?”, (**c**) “What do you know about them?”, (**d**) “What are they doing when you see them?”.

**Table 1 animals-13-03285-t001:** Naming task answer distribution displaying the percentage of correct, incorrect, and “unknown” responses for 13 non-loris species. For incorrect identifications, misidentifications are listed.

Species Name	Correct (*%*)	Incorrect (*%*)	Unknown (*%*)	Misidentification(s)
Malayan colugo (*Galeopterus variegatus)*	67	11	22	“bat” (N = 3), “flying squirrel” (N = 1)
Fishing cat (*Prionailurus viverrinus)*	61	8	31	“civet” (N = 2), “leopard” (N = 1)
Masked palm civet (*Paguma larvata*)	47	3	50	“wild cat” (N = 1)
Nicobar pigeon (*Caloenas nicobarica*)	8	8	83	“green bird” (N = 1), “peacock” (N = 1), “red turtle dove” (N = 1)
Water monitor (*Varanus salvator*)	100	-	-	*-*
Tokay gecko (*Gekko gecko*)	100	-	-	*-*
Southern tree shrew (*Tupaia glis*)	97	-	3	-
Sunda pangolin (*Manis javanica*)	94	-	6	-
Indian muntjac (*Muntiacus muntjak*)	42	-	58	-
Common birdwing (*Troides helena*)	31	-	69	-
Pied imperial pigeon (*Ducula bicolor*)	8	-	92	-
Common brown lemur (*Eulemur fulvus*)	-	19	81	“squirrel” (N = 2), “civet” (N = 2), “loris” (N = 1), “raccoon” (N = 1), “langur” (N = 1)
Blue monkey (*Cercopithecus mitis*)	-	6	94	“howling monkey” (N = 1), “long-tailed macaque” (N = 1)

**Table 2 animals-13-03285-t002:** Represents the 11 key concepts that define the cultural domain for the 36 interviews that we conducted in Khao Lak, Thailand. The examples are translated quotes from informants and the numbers represent how many interviews in which these concepts appeared and their percentage out of the total number of interviews.

Concept	Example	Number
1. Awake at night	“They can only be seen at night”	25 (69%)
2. Eat fruits	“They eat the fruit in front of the house and that is when we see them”	15 (42%)
3. We do not eat/hunt	“We do not hunt them or eat them, but we don’t know why”	12 (33%)
4. Bad omen	“The old people always say that they are a bad omen…We don’t know why it is a bad omen”	9 (25%)
5. Used to be more/see less	“There used to be a lot more, even in town”	9 (25%)
6. Not aggressive	“They are not aggressive if you don’t touch them”	9 (25%)
7. Shoo from house	“It’s not a popular animal, people don’t like them that much because if it goes to a house, it means that something bad is happening to the people there…Usually, people shoo them out of the house”	8 (22%)
8. They bite	“You can’t catch them because they bite”	7 (19%)
9. Electrocuted	“They walk around on the wires and get electrocuted, and one died in front of our house a couple of days ago”	6 (16%)
10. We do not touch	“Because of superstition, it is bad to touch them”	5 (14%)
11. Eat insects	“They eat fruit and red ants. Last night, it went to a mango tree by my house to eat fruit and red ants”	5 (14%)

## Data Availability

The data presented in this study are available upon request from the corresponding author. The data are not publicly available due to privacy reasons.

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
