# Peer review of "Knowledge, Beliefs, and Experience Regarding Slow Lorises in Southern Thailand: Coexistence in a Developed Landscape"

_animals, 2023, doi:10.3390/ani13203285_

Round 1
Reviewer 1 Report
The authors of this study conducted 36 interviews with local residents in southern Thailand's Khao Lak region to investigate their knowledge, beliefs, and experiences related to slow lorises, shedding light on beliefs around Lorisis, their interactions with humans, and threats in the area. Research on primate-human interactions in deforested, human-dominated areas can be really useful for conservation by revealing human-wildlife dynamics and the value of local knowledge. This knowledge is useful for creating evidence-backed strategies for future conservation related research and to foster human-wildlife coexistence. Slow lorises were observed in both rural and urban areas, indicating their adaptability to human-dominated environments. It also revealed that people in the Khao Lak region generally coexist peacefully with slow lorises, with minimal conflict and exploitation and suggested that electrocutions and road accidents may be primary human related threats to slow lorises in this area.
Strengths of the Study
This was a well written and polished manuscript and the interpretations of results seemed reasonable to me. I have no comments or suggestions on the writing. One of the notable strengths of the study is its use of local knowledge through interviews with residents. This approach provides valuable insights into the perceptions and experiences of local communities, which can be instrumental in understanding the human-wildlife dynamics in the study area. The study's focus on nocturnal and cryptic species like slow lorises is noteworthy. Such species often receive less attention in ecological research, making this study a valuable contribution to understanding their ecology and interactions with human-dominated environments. The study's findings have some obvious practical implications for conservation action plans. It goes beyond typical ecological research by identifying threats and areas where local knowledge can inform conservation efforts.
Limitations of the Study:
One potential limitation of the study is the relatively small sample size. Interview studies are not something I have done, so cannot state if this type of sample size is common in these types of studies. Regardless, I feel the study authors should acknowledge the limitations of the relatively small sample size- ( 36 interviews, which may not represent the broader population's perspectives and experiences)- while also explaining why the results still are relevant and represent an important contribution. Likewise it may also be good to discuss briefly the limitations of in person interviews when respondents may give biased answers to strangers- if you did anything to mitigate this, and discuss this in context of beneficial knowledge gained.
I can not comment on the statistical methods, with which I am not familiar.
Author Response
Thank you for your kind consideration of our manuscript. We are grateful for the detailed comments provided to us and feel that after careful incorporation of this feedback we have made significant improvement to our work. We have addressed your comments and concerns on line 130, 159 and additionally at 420 with a paragraph exploring limitations and how they were addressed.

Reviewer 2 Report
This is an interesting paper concerning species where there is a gap in knowledge. Authors have conducted an exploratory qualitative study that is enriching and relevant to the species and the region. The paper is well referenced and much of the discussion is supported by compelling evidence from other studies. They display extensive understanding of the species and context. Their findings are well conveyed, with clear key recommendations that are relevant for conservation action and further investigation.
However, I believe there is a lack of rigor in some of the analysis and discussion as some assumptions are made without enough data to sustain these. My concern refers to the design of the questions (see comments for lines 284, 305 and 417). Furthermore, no questions refer directly to attitudes or behaviors towards the lorises, and these are only inferred by the authors as a result of participants answers to the question 'What do you know about them?', which is very general. If participants were not asked about 'what people do when they meet the loris' it is difficult to make assumptions about loris-human interactions.
Because the data may be useful to forward the understanding of the species and to inform further conservation actions, I recommend that the authors acknowledge the limitations caused by the research design, reiterate assumption that are not sufficiently supported by responses and refer to these as 'suggestions', and areas of interest for further investigation.
line 186: What is the total number of respondents who recognised the loris (and were subsequently interviewed), and the total number of people approached? This ratio provides more information on local context and familiarity with the loris species.
188: this data does appear in the results or discussion. Please explain what you learned from it, and why it was not used in this paper.
212: Please provide the total number N=36?
233: please revise this sentence: "Represents the 11 key concepts that define the cultural domain for the 36 slow loris (Nycticebus spp.) Four interviews that we conducted in Khao Lak, Thailand"
284: I am not sure about this inference since none of the questions listed refers directly to exploitation or harm. If you argue that participants answers did not suggest exploitation or conflict, you may want to recommend further investigation focused on exploitation and harm in particular.
305: If lorises are protected and killing them is a crime, people are expected to be resistant to disclose any damaging behaviour towards the animals. One way of bypassing such limitation in research design would be to ask people about 'what others do' (not themselves in particular). I recommend the authors acknowledge the limitations in research design and suggest that in the future further investigation is carried out to address any fear of retaliation from authorities on the part of participants.
410: As a result of this study, areas for further investigation have been identified. Including the key areas for further investigation identified in Discussion would improve the scope of the Conclusions.
417: this may need to be further investigated using different research questions, as suggested above. Data on loris population numbers and trends may also help support or challenge this assumption.
Author Response
Thank you for your kind consideration of our manuscript. We are grateful for the detailed comments provided to us and feel that after careful incorporation of this feedback we have made significant improvement to our work. We worked to address all of the comments and have detailed the changes in the attached word document.

Round 2
Reviewer 2 Report
Thank you for addressing all areas that I have raised in my comments. I am glad you have found these useful. I believe my concerns and suggestions have been addressed adequately and in full and the manuscript is now ready for publication.